# Law of Total Probability in Quantum Theory and Its Application in Wigner’s Friend Scenario

**DOI:** 10.3390/e24070903

**Published:** 2022-06-29

**Authors:** Jianhao M. Yang

**Affiliations:** Qualcomm, San Diego, CA 92121, USA; jianhao.yang@alumni.utoronto.ca

**Keywords:** the law of total probability, extended Wigner’s friend scenario, PVOM measurement

## Abstract

It is well-known that the law of total probability does not generally hold in quantum theory. However, recent arguments on some of the fundamental assumptions in quantum theory based on the extended Wigner’s friend scenario show a need to clarify how the law of total probability should be formulated in quantum theory and under what conditions it still holds. In this work, the definition of conditional probability in quantum theory is extended to POVM measurements. A rule to assign two-time conditional probability is proposed for incompatible POVM operators, which leads to a more general and precise formulation of the law of total probability. Sufficient conditions under which the law of total probability holds are identified. Applying the theory developed here to analyze several quantum no-go theorems related to the extended Wigner’s friend scenario reveals logical loopholes in these no-go theorems. The loopholes exist as a consequence of taking for granted the validity of the law of total probability without verifying the sufficient conditions. Consequently, the contradictions in these no-go theorems only reconfirm the invalidity of the law of total probability in quantum theory rather than invalidating the physical statements that the no-go theorems attempt to refute.

## 1. Introduction

In his seminal paper on the path integral formulation of quantum mechanics [1], Feynman started the introduction of his new theory by pointing out that the law of total probability in classical probability theory must be replaced by a new form of rule. Specifically, in a slightly different notation, the classical law of probability, $p\left(c|a\right)=\sum_{b}p\left(c|b\right)p\left(b|a\right)$ where p(y|x) is the probability of obtaining measurement result *y* given measurement result *x*, is no longer true in quantum theory and must be replaced by $\varphi \left(c|a\right)=\sum_{b}\varphi \left(c|b\right)\varphi \left(b|a\right),$ where φ is a complex number called probability amplitude and related to classical probability by Born’s rule $p\left(y|x\right)={\left|\varphi \left(y|x\right)\right|}^{2}$. From this key idea, Feynman continued to expand the theory that led to the path integral formulation of quantum mechanics. He also discussed when the new rule of summation over probability amplitude can fall back to the classical law of probability. This is when one “attempts to perform” intermediate measurements that obtain results of all *b*. In modern terms, what Feynman means by “attempting to perform measurement” can be understood as the decoherence phenomenon [2].

The above example shows that it has been long known that the law of total probability cannot be taken for granted in quantum theory. Indeed, many other classical probability rules are only upheld in specific conditions. For instance, a joint probability can be definitely assigned only when the two measurement operators are commutative [3,4,5,6]. There are many variants of definitions of the conditional probability in quantum theory (for a review, see [7]). However, a family of no-go theorems recently published [8,9,10,11] appears to rely on the total law of probability one way or another without considering the sufficient conditions. These no-go theorems are related to the extensively discussed Wigner’s friend experiments. In quantum mechanics, the Wigner’s friend [12,13] thought experiment has been widely discussed, as it tests the validity of many quantum interpretation theories. The significance of such experiments is that Wigner and his friend give two different descriptions of the same physical process happening inside the lab. Deutsch further extended the thought experiment to be applicable to macroscopic system such as the lab system [14] itself. Based on that, a more sophisticated extended Wigner’s friend experiment is put forwarded by Brukner [8,15]. Such an experimental setup involves two remotely separated labs. Each lab contains half of an entangled pair of spins and a local observer. Outside each lab there is a super-observer who can choose to perform different types of measurements on the lab as a whole. The intention of such an experimental setup is to prove, through a no-go theorem, that measured facts are observer-dependent in quantum theory. A subsequent experiment [16] has been carried out to confirm the inequality developed in [8]. A stronger version of the no-go theorem is further proposed for reaching a similar conclusion [9]. The statement that measured facts are observer-dependent was considered important for the quantum foundation and deserved rigorous theoretical proving and experimental testing. However, proving the no-go theorems by taking the law of total probability for granted casts doubt on their theoretical rigorousness.

The fact that there is still ambiguity in using the total law of probability in quantum theory—though it has long been recognized as not being upheld in quantum mechanics—shows the need to provide a rigorous formulation of the law of total probability in quantum theory and to clarify under what conditions it holds true. This is indeed the motivation behind the present work. Formulation of the law of total probability depends on a clear definition of conditional probability in quantum theory. There is already extensive research on how conditional probability is defined [7,17,18,19,20,21,22,23,24,25]. However, these formulations are either based on projection measurements or only consider simultaneous measurements with commutative operators. In this work, I extend a two-time conditional probability formulation from projection measurement to more generic POVM measurements. Generalization for POVM measurement is needed because some of the no-go theorems choose POVM operators in their proofs. I then give several sufficient conditions for the law of total probability to become true. The theory is applied to analyze several no-go theorems related to the extended Wigner’s friend scenario. Logical loopholes are shown in these no-go theorems because their proofs rely on the law of total probability one way or another, but the conditions to validate the law are not met. Thus, these no-go theorems do not really prove the results they expect, such as “measured facts are observer-dependent”. Instead, they just indirectly confirm that the law of total probability does not not hold in quantum theory.

It is worth mentioning that other concerns regarding these no-go theorems have already been pointed out [26,27]. In particular, only when a measurement is completed should a probability distribution be assigned. Assigning probability distribution for pre-measurement without results leads to contradiction [26]. The analysis in this work will go one step further by showing that even assigning a probability distribution for completed measurements still leaves logical loopholes in the no-go theorem. This is because the law of total probability that the proofs rely on does not hold true with the specific measurement operators and initial quantum state being chosen. Lastly, it is important to emphasize that I do not take a stand on the assertions of the no-go theorems themselves. For instance, it could still be a valid statement that “measured facts are observer-dependent”. What I only show here is that there are logical loopholes in the proof of the no-go theorems.

In summary, this paper extends the formulation of conditional probability to generic POVM measurements and clarifies the conditions under which the law of total probability can be valid in quantum theory. Applying the theory developed in this work to the extended Wigner friend scenario reveals logical loopholes in several no-go theorems that take for granted the validity of the law of total probability. The contradictions in these no-go theorems only reconfirm the invalidity of the law of total probability in quantum theorem rather than invalidating the physical statements that the no-go theorems are intended to refute, such as “measured facts are independent of the observer”. I hope the results presented here inspire further research to find more convincing proof and experimental testing. This is important because the implications of the extended Wigner’s friend scenario are conceptually fundamental in quantum theory.

## 2. The Law of Total Probability in Quantum Theory

First, I briefly review classical probability theory. Suppose there are two random variables X and Y. Without loss of generality, I assume X and Y are discrete random variables. Measuring *X* (or *Y*) will obtain one of the values in $\left\{a_{i}:i=1,2,3\ldots \right\}$ (or in $\left\{b_{j}:j=1,2,3\ldots \right\}$), which is finite or countable infinite. Denote the joint probability of measuring X with result $X=a_{i}$, measuring Y with result $Y=b_{j}$ as $p(a_{i},b_{j})$, and the conditional probability of obtaining $X=a_{i}$ given that $Y=b_{j}$ as $p(a_{i}|b_{j})$. They are related by the following axioms: (1)$$\begin{array}{cc} p(a_{i},b_{j}) & =p\left(b_{j}|a_{i}\right)p\left(a_{i}\right) \end{array}$$(2)$$\begin{array}{cc} p(b_{j},a_{i}) & =p\left(a_{i}|b_{j}\right)p\left(b_{j}\right) \end{array}$$(3)$$\begin{array}{cc} p(a_{i},b_{j}) & =p(b_{j},a_{i}), \end{array}$$
where $p(a_{i})$ is the marginal probability of measuring X with result $X=a_{i}$, and similarly for $p(b_{j})$. Axiom (3) ensures the joint probability is defined uniquely regardless if it is defined by (1) or (2). We explicitly call out (3) since it is not always true in quantum theory.

The law of total probability can be derived (Axioms (1)–(3) give $p\left(b_{j}|a_{i}\right)p\left(a_{i}\right)=p\left(a_{i}|b_{j}\right)p\left(b_{j}\right)$, which is Bayes’ law. Summing over *i* on both sides and using identity $\sum_{i}p\left(a_{i}|b_{j}\right)p\left(b_{j}\right)=p\left(b_{j}\right)$, one obtains (4)) from axioms (1)–(3), expressed as following,
(4)$$p\left(b_{j}\right)=\sum_{i}p\left(b_{j}|a_{i}\right)p\left(a_{i}\right).$$
What I want to investigate here is how the equivalent version of (4) in quantum theory can be formulated.

To start with, I need to examine how conditional probability is constructed in quantum theory. The subtlety of constructing conditional probability in quantum theory has been investigated long ago. G. Bobo gives an extensive review and discussion [7]. The generally accepted formulation of conditional probability in quantum theory is provided by Lüders rule [18], where the measurements are associated with projection operators. Lüders rule is based on Gleason’s theorem, which mathematically justifies Born’s rule. Here I wish to follow a similar approach to generalize the formulation for conditional probability when the measurements are associated with POVM operators.

Mathematical proofs for generalizing Gleason’s theorem to POVM measurements are given by [28,29], which is our starting point. Suppose a quantum system *S* is prepared such that its state is described by density operator ρ. *S* could be a composite system, which I will discuss later. Let $A=\{A_{i}\}$ be a POVM for *S*. The probability of measurement with element $A_{i}$ resulting in value $a_{i}$ is [28,29].
(5)$$p\left(a_{i}|\rho \right)=Tr\left(\rho A_{i}\right),$$
and the post-measurement density operators $\rho_{i}$ are given by [4]
(6)$$\rho_{i}=\frac{\sqrt{A_{i}}\rho \sqrt{A_{i}}}{p(a_{i}|\rho )}.$$
Let $B=\{B_{j}\}$ be another POVM for *S*. Given post measurement state $\rho_{i}$, the probability of measurement with element $B_{j}$ resulting in value $b_{j}$ is, by recursively applying (5), $p\left(b_{j}|\rho_{i}\right)=Tr\left(\rho_{i}B_{j}\right)$. Substituting the expression for $\rho_{i}$ in (6), I obtain the conditional probability
(7)$$p\left(b_{j}|a_{i},\rho \right)=\frac{Tr(\sqrt{A_{i}}\rho \sqrt{A_{i}}B_{j})}{Tr(\rho A_{i})}.$$
There is an underlying assumption in this definition that the probability is assigned only after the measurements are completed. In particular, the first POVM measurement $A_{i}$ must be completed in order to be qualified as a condition. We strictly follow this assumption as opposed to assigning a probability with only “pre-measurement”. Pre-measurement refers only to the unitary process that entangles the measured system and measuring apparatus [30] but without the projection process to single out a particular outcome.

Given the same initial state ρ, if I swap the order of measurements such that $B_{j}$ goes first, followed by $A_{i}$, I obtain a conditional probability
(8)$$p\left(a_{i}|b_{j},\rho \right)=\frac{Tr(\sqrt{B_{j}}\rho \sqrt{B_{j}}A_{i})}{Tr(\rho B_{j})}.$$
Note $p\left(a_{i}|\rho \right)=Tr\left(\rho A_{i}\right)$ and $p\left(b_{i}|\rho \right)=Tr\left(\rho B_{j}\right)$; Equations (7) and (8) can be rewritten as
(9)$$\begin{array}{cc} p\left(b_{j}|a_{i},\rho \right)p\left(a_{i}|\rho \right) & =Tr(\sqrt{A_{i}}\rho \sqrt{A_{i}}B_{j}) \end{array}$$
(10)$$\begin{array}{cc} p\left(a_{i}|b_{j},\rho \right)p\left(b_{j}|\rho \right) & =Tr(\sqrt{B_{j}}\rho \sqrt{B_{j}}A_{i}). \end{array}$$
Equations (9) and (10) are not necessarily equal, which indicate that the quantum version of Bayes’ theorem,
(11)$$p\left(a_{i}|b_{j},\rho \right)p\left(b_{j}|\rho \right)=p\left(b_{j}|a_{i},\rho \right)p\left(a_{i}|\rho \right)$$
does not hold in general in quantum theory. This posts a difficulty to define a joint probability as either $p\left(a_{i},b_{j}\right)=p\left(a_{i}|b_{j},\rho \right)p\left(b_{j}|\rho \right)$ or $p\left(a_{i},b_{j}\right)=p\left(b_{j}|a_{i},\rho \right)p\left(a_{i}|\rho \right)$ because it depends on the order of measurement events. Another consequence is that the laws of total probability, i.e., the quantum version of (4)
(12)$$\sum_{i}p\left(b_{j}|a_{i},\rho \right)p\left(a_{i}|\rho \right)=p\left(b_{j}|\rho \right)$$
does not hold in general either. This is because from (9), $\sum_{i}p\left(b_{j}|a_{i},\rho \right)p\left(a_{i}|\rho \right)=\sum_{i}Tr$$\left(\sqrt{A_{i}}\rho \sqrt{A_{i}}B_{j}\right)$, while $p\left(b_{j}|\rho \right)=Tr\left(\rho B_{j}\right)$, and these are not equal in general (Note that on the other hand, given (7) and the completeness of POVM elements, $\sum_{i}A_{i}=I$, where *I* is the identity operator, it is straightforward to verify that $\sum_{j}p\left(b_{j}|a_{i},\rho \right)p\left(a_{i}|\rho \right)=p\left(a_{i}|\rho \right)$). We are interested in finding the conditions under which (12) becomes true.

It is well-known that when $[A_{i},B_{j}]=0$, i.e., $A_{i}$ and $B_{j}$ commute, from (7) and (8), one gets $p\left(b_{j}|a_{i},\rho \right)p\left(a_{i}|\rho \right)=p\left(a_{i}|b_{j},\rho \right)p\left(b_{j}|\rho \right)=Tr\left(\rho A_{i}B_{j}\right)$. Consequently, the law of total probability (12) becomes true and a joint probability can be well-defined. However, the situation becomes much complicated when $[A_{i},B_{j}]\neq 0$.

Strictly speaking, due to the uncertainty principle, when $A_{i}$ and $B_{j}$ are non-commutative, the two measurements cannot be performed to obtain definite outcomes at the same time. The conditional probability defined in (7) or (8) needs to be extended to a two-time formulation of conditional probability in order to be applicable when $[A_{i},B_{j}]\neq 0$. There is extensive research on how to construct two-time conditional probability in quantum theory [7,17,18,19,20,21,22,23,24,25]. One noticeable approach is based on the Page–Wootters timeless formulation [21,22,23,24,25]. However, this work will continue to be based on the generalized Gleason theorem for POVM [28,29] to derive the two-time conditional probability, and will leave discussion of the Page–Wootters mechanism for Section 4.

For conceptual clarity, I start the analysis by considering that there is finite nonzero duration for each measurement. After I construct the conditional probability formulation, for practical purpose of calculation, I can approximate the measurement duration to zero. Suppose the first measurement starts at $t_{a}^{-}$ and completes at $t_{a}^{+}$. Here $t_{a}^{+}-t_{a}^{-}$ covers the time duration for both the pre-measurement unitary phase that entangles the measured system and the measuring apparatus, and the projection phase. The measurement process (Theorem 5.2 of [4] gives a detailed account on how this POVM measurement is physically realized through indirect measurement) is represented by a POVM element $A_{i}$ associated with outcome $a_{i}$. Similarly, the second measurement starts at $t_{b}^{-}$ and completes at $t_{b}^{+}$. Between $t_{a}^{+}$ and $t_{b}^{-}$ there is a free time evolution for the measured system *S*, described by operator $U\left(\Delta t\right)=e^{-iH\Delta t/\hbar }$, where $\Delta t=(t_{b}^{-}-t_{a}^{+})$. Since it is only meaningful to assign a probability distribution after a measurement is completed, the two-time conditional probability I want to construct is “given the measurement outcome of $a_{i}$ at $t_{1}$ where $t_{a}^{+}<t_{1}<t_{b}^{-}$, what is the probability of measurement outcome *b* at $t_{2}>t_{b}^{+}$". Mathematically, this two-time conditional probability can be written as $p(b_{j}\;\mathrm{at}\;t_{2}|a_{i}\;\mathrm{at}\;t_{1},\rho_{0})$, where $\rho_{0}\equiv \rho \left(t_{a}^{-}\right)$ is the initial density operator of *S* when the first measurement starts. After the first measurement with POVM element $A_{i}$, the post-measurement state is $\rho_{i}\left(t_{a}^{+}\right)=\sqrt{A_{i}}\rho_{0}\sqrt{A_{i}}/Tr\left(\rho_{0}A_{i}\right)$. The quantum system *S* then time evolves from $t_{a}^{+}$ to $t_{b}^{-}$ to a new state $\rho_{i}\left(t_{b}^{-}\right)=U\left(\Delta t\right)\rho_{i}\left(t_{a}^{+}\right)U^{†}\left(\Delta t\right)$. At $t_{b}^{-}$, the second measurement occurs. This is represented by applying POVM element $B_{j}$ on $\rho_{i}\left(t_{b}^{-}\right)$ and obtaining outcome $b_{j}$ at $t_{b}^{+}$ with probability $Tr(B_{j}\rho_{i}\left(t_{b}^{-}\right))$. Substituting $\rho_{i}\left(t_{b}^{-}\right)$, the two-time conditional probability is
(13)$$\begin{array}{cc} \; & p\left(b_{j}\;\mathrm{at}\;t_{2}|a_{i}\;\mathrm{at}\;t_{1},\rho_{0}\right)=Tr\left(B_{j}\rho_{i}\left(t_{b}^{-}\right)\right)\\ \; & =\frac{Tr(B_{j}U\left(\Delta t\right)\sqrt{A_{i}}\rho_{0}\sqrt{A_{i}}U^{†}\left(\Delta t\right))}{Tr(\rho_{0}A_{i})}. \end{array}$$

For practical purposes of calculation, I can assume the measurement duration is very small compared to the free evolution time, i.e., $(t_{a}^{+}-t_{a}^{-})\ll \Delta t$ and $(t_{b}^{+}-t_{b}^{-})\ll \Delta t$. Then, I can denote $t_{a}^{-}\approx t_{a}^{+}$ as $t_{a}$, $t_{b}^{-}\approx t_{b}^{+}$ as $t_{b}$, and $\Delta t=(t_{b}-t_{a})$.

Suppose the two POVM elements $A_{i}$ and $B_{j}$ are projection measurements, $A_{i}=|\phi_{i}\rangle \langle \phi_{i}|$ and $B_{j}=|\varphi_{j}\rangle \langle \varphi_{j}|$; one can verify that the conditional probability defined in (13) gives the correct transition probability in standard quantum mechanics:(14)$$p\left(b_{j}\;\mathrm{at}\;t_{2}|a_{i}\;\mathrm{at}\;t_{1},\rho_{0}\right)=|\langle \phi_{i}|U\left(\Delta t\right){\left|\varphi_{j}\right|}^{2}.$$
However, Equation (13) is more generic as it is defined with general POVM operators. Note that the denominator in (13) $Tr\left(\rho_{0}A_{i}\right)=p\left(a_{i}\;\mathrm{at}\;t_{1}|\rho_{0}\right)$; Equation (13) can be rewritten as
(15)$$\begin{array}{cc} \; & p\left(b_{j}\;\mathrm{at}\;t_{2}|a_{i}\;\mathrm{at}\;t_{1},\rho_{0}\right)p\left(a_{i}\;\mathrm{at}\;t_{1}|\rho_{0}\right)\\ \; & =Tr(B_{j}U\left(\Delta t\right)\sqrt{A_{i}}\rho_{0}\sqrt{A_{i}}U^{†}\left(\Delta t\right)). \end{array}$$
To analyze the two-time version of the total law of probability, which can be expressed as
(16)$$\begin{array}{cc} p(b_{j}\;\mathrm{at}\;t_{2}|\rho_{0})\; & =\sum_{i}p\left(b_{j}\;\mathrm{at}\;t_{2}|a_{i}\;\mathrm{at}\;t_{1},\rho_{0}\right)p\left(a_{i}\;\mathrm{at}\;t_{1}|\rho_{0}\right), \end{array}$$
I consider a series of two-time measurements $\left\{A_{i}\;\mathrm{at}\;t_{a},B\;\mathrm{at}\;t_{b},i=1\ldots N\right\}$ on *N* copies of measured system *S* with the same initial state $\rho_{0}$. Each two-time measurement consists a first measurement from one possible POVM element from the complete set $\left\{A_{i},i=1\ldots N\right\}$ at time $t_{a}$ and the same second measurement $B_{j}$ at time $t_{b}$. For $t_{a}<t_{1}<t_{b}<t_{2}$, from (15) I have
(17)$$\begin{array}{cc} \; & \sum_{i}p\left(b_{j}\;\mathrm{at}\;t_{2}|a_{i}\;\mathrm{at}\;t_{1},\rho_{0}\right)p\left(a_{i}\;\mathrm{at}\;t_{1}|\rho_{0}\right)\\ \; & =\sum_{i}Tr\left(B_{j}U\left(\Delta t\right)\sqrt{A_{i}}\rho_{0}\sqrt{A_{i}}U^{†}\left(\Delta t\right)\right). \end{array}$$
However, by definition, $p\left(b_{j}\;\mathrm{at}\;t_{2}|\rho_{0}\right)=Tr\left(B_{j}U\rho_{0}U^{†}\right)$. We can see (16) is not true in general. The Theorem next attempts to address the question of under what conditions (16) is valid.

**Theorem** **1.** 
*Let $\rho_{0}$ be the density operator for a quantum system S before the measurements. Let $A_{i}$ and $B_{j}$ be two POVM elements to measure S at time $t_{a}$ and $t_{b}$, respectively, and $U(t_{b},t_{a})$ is the unitary time evolution operator from $t_{a}$ to $t_{b}$. Select $t_{1}$ and $t_{2}$ such that $t_{a}<t_{1}<t_{b}<t_{2}$. The law of total probability (16) is true if one of the following conditions is met.*
*C1.* 
*$[A_{i},U^{†}B_{j}U]=0$, $\forall \rho_{0}$,*
*C2.* 
*$\rho_{0}=\sum_{i}\lambda_{i}|\phi_{i}\rangle \langle \phi_{i}|$ and $A_{i}=|\phi_{i}\rangle \langle \phi_{i}|$,*
*C3.* 
*$\rho_{0}$ is a pure state, given by |Ψ〉〈Ψ|, $A_{i}$ is a projection operator and $\langle \Psi |\left[A_{i},U^{†}B_{j}U\right]A_{i}|\Psi \rangle =0$.*



The proof of Theorem 1 is in Appendix A, but a few comments are in order here. First, Condition C1 implies $U^{†}B_{j}UA_{i}=A_{i}U^{†}B_{j}U$. The sequence of operations for $U^{†}B_{j}UA_{i}$ means performing measurement $A_{i}$ at $t_{a}$, time evolving the post-measurement state from $t_{a}$ to $t_{b}$, performing measurement $B_{j}$ at $t_{b}$, and reversing time evolution of the post-measurement state back to $t_{a}$. The sequence of operations $A_{i}U^{†}B_{j}U$ means time evolving the state from $t_{a}$ to $t_{b}$, performing measurement $B_{j}$ at $t_{b}$, then reversing time evolution of the state back to $t_{a}$, and performing measurement $A_{i}$ at $t_{a}$. Condition C1 says that if these two sequences of operations are equivalent, then the law of total probability (16) holds true.

Second, if the post-measurement state $\rho_{i}\left(t_{a}\right)$ after the first measurement does not change during free time evolution, such as the case of a spin state in free space, one will have $\rho_{i}\left(t_{b}\right)=\rho_{i}\left(t_{a}\right)=\sqrt{A_{i}}\rho_{0}\sqrt{A_{i}}/Tr\left(\rho_{0}A_{i}\right)$. Then, Equation (13) can be written as
(18)$$p\left(b_{j}\;\mathrm{at}\;t_{2}|a_{i}\;\mathrm{at}\;t_{1},\rho \right)=\frac{Tr(\rho_{0}\sqrt{A_{i}}B_{j}\sqrt{A_{i}})}{Tr(\rho_{0}A_{i})}.$$
Equation (18) appears the same as (7), but the precise meaning is different in that the two measurements $A_{i}$ and $B_{j}$ in (18) are taken at two different times. With such a special post-measurement quantum state, the sufficient conditions in Theorem 1 become

*C1′.* 
*$[A_{i},B_{j}]=0$, $\forall \rho_{0}$,*
*C2′.* 
*$\rho_{0}=\sum_{i}\lambda_{i}|\phi_{i}\rangle \langle \phi_{i}|$ and $A_{i}=|\phi_{i}\rangle \langle \phi_{i}|$,*
*C3′.* 
*$\rho_{0}=\left|\Psi \rangle \langle \Psi \right|$, $A_{i}$ is a projection operator and $\langle \Psi |\left[A_{i},B_{j}\right]A_{i}|\Psi \rangle =0$.*


A couple of comments are in order before closing this section. First, when two measurement operations are not commutative, the conditional probability needs to be defined in the two-time formulation. Second, I can give an intuitive explanation of why (16) does not hold in general in quantum theory. As shown in (17), the right-hand side of (16) refers to the summation of traces of multiplication of operators from a series of experiments where two measurements are carried out in a sequence. In the case of a special post-measurement state where (18) holds, this is $\sum_{i}Tr\left(B_{j}\sqrt{A_{i}}\rho_{0}\sqrt{A_{i}}\right)$. Measurement of $A_{i}$ changes the initial quantum state such that it affects the probability of outcome for a subsequent measurement $B_{j}$. However, the term on the left-hand side of (16) refers to the probability of an experiment where only measurement $B_{j}$ is carried out with the same initial quantum state. There is no reason to assume both sides are equal. Equation (16) holds only in special conditions such as those specified in Theorem 1.

The conclusion here is that one should not take for granted that the law of total probability holds true in general. Instead, sufficient conditions, such as those provided in Theorem 1, need to be clearly called out. Failing to do so may leave a loophole in logical deduction when applying the law of total probability.

## 3. Application to Composite Systems

In this subsection, I will apply the conditional probability definition to composite quantum systems and reexamine Theorem 1 when measuring composite systems. Suppose the measured system *S* consists of two subsystems $S_{1}$ and $S_{2}$ that are space-like separated. Define $A_{i}=P_{i}⊗I_{2}$, where $P_{i}=|\phi_{i}\rangle \langle \phi_{i}|$ is a local POVM element on subsystem $S_{1}$, and $I_{2}$ is an identity operator on subsystem $S_{2}$. Similarly, define $B_{j}=I_{1}⊗Q_{j}$, where $Q_{j}$ is a local POVM element on subsystem $S_{2}$. By the principle of locality, a local measurement on a subsystem should not impact the other remote subsystem. Therefore, $[A_{i},B_{j}]=0$. For measurement outcomes of two such local measurements, Equations (7) and (8) are correct formulations for conditional probability; the joint probability is well-defined. Consequently, Equations (11) and (12) hold true. There is no need to use the two-time formulation of conditional probability. This is the case for typical Bell tests and has been used to derive the Bell–CHSH inequalities (On the other hand, in the derivation of Bell–CHSH inequalities, identity (1) is further expressed as
(19)$$p\left(a_{i},b_{j}|\lambda \right)=p\left(a_{i}|b_{j},\lambda \right)p\left(b_{j}|\lambda \right)=p\left(a_{i}|\lambda \right)p\left(b_{j}|\lambda \right),$$
where λ is a hidden variable. This is known as the *outcome independence* assumption [31,32]).

However, suppose $B_{j}=Q_{j}⊗I_{2}$, where $Q_{j}$ is another local POVM element on subsystem $S_{1}$, and $[P_{i},Q_{j}]\neq 0$. In this case, Equation (7) is incorrect for conditional probability. The two-time conditional probability formulation is needed and can be calculated as
(20)$$\begin{array}{cc} \; & p(b_{j}\;\mathrm{at}\;t_{2}|a_{i}\;\mathrm{at}\;t_{1},\rho_{0})\\ \; & =\frac{Tr(\left(Q_{j}⊗I_{2}\right)U\left(\Delta t\right)\sqrt{P_{i}⊗I_{2}}\rho_{0}\sqrt{P_{i}⊗I_{2}}U^{†}\left(\Delta t\right))}{Tr(\rho_{0}P_{i}⊗I_{2})}, \end{array}$$
where $U\left(\Delta t\right)=U_{S_{1}}\left(\Delta t\right)⊗U_{S_{2}}\left(\Delta t\right)$.

Next, I wish to apply the two-time conditional probability to the extended Wigner’s friend (EWF) scenario introduced in [8]. As shown in Figure 1, the EWF scenario consists of two space-like separated laboratories $L_{1}$ and $L_{2}$. Each laboratory contains half of an entangled pair of systems $s_{1}$ and $s_{2}$. $L_{1}$ also contains a friend Charlie who can perform measurements on $s_{1}$. Outside $L_{1}$ there is a super-observer Alice who can perform different types of measurements on $L_{1}$ as a whole. Similarly, there is a friend Debbie in $L_{2}$ and a super-observer Bob outside $L_{2}$. Here, four POVM measurements are needed and represented by POVM elements A,B,C,D, where operators *A* and *C* act on Hilbert space $H_{L_{1}}$, and *B* and *D* act on Hilbert space $H_{L_{2}}$. I drop the subscripts of the operators and $\rho_{0}$ for simplifying notations. In a typical EWF experiment, the chosen operators are not all commutative with one another. Specifically, [A,C]≠0 and [B,D]≠0, while $[C⊗I_{L_{2}},I_{L_{1}}⊗D]=0$ and $[A⊗I_{L_{2}},I_{L_{1}}⊗B]=0$. The two-time probability formulation to compute the conditional probability is needed because measurements *C* and *D* are taken before measurements *A* and *B*. Since $[C⊗I_{L_{2}},I_{L_{1}}⊗D]=0$ and $[A⊗I_{L_{2}},I_{L_{1}}⊗B]=0$, I can assume measurements *C* and *D* are taken at the same time, $t_{a}$, as C⊗D, while measurement *A* and *B* are taken at the same, later time $t_{b}$ as A⊗B. Without loss of clarity, I drop the symbol ⊗ hereafter. Then, the conditional probability for $t_{a}<t_{1}<t_{b}<t_{2}$ is given by
(21)$$\begin{array}{cc} \; & p(ab\;\mathrm{at}\;t_{2}|cd\;\mathrm{at}\;t_{1},\rho ,xy)\\ & =\frac{Tr(\rho \sqrt{C}\sqrt{D}U^{†}ABU\sqrt{C}\sqrt{D})}{Tr(\rho CD)}, \end{array}$$
where $U=U_{L_{1}}⊗U_{L_{2}}$ is the time evolution operator from $t_{a}$ to $t_{b}$. The law of total probability I am interested in is
(22)$$\begin{array}{cc} \; & p(ab\;\mathrm{at}\;t_{2}|\rho ,xy)=\\ \; & \sum_{cd}p\left(ab\;\mathrm{at}\;t_{2}|cd\;\mathrm{at}\;t_{1},\rho ,xy\right)p\left(cd\;\mathrm{at}\;t_{1}|\rho ,xy\right). \end{array}$$
From (21), the R.H.S. of (22) becomes
(23)$$\begin{array}{cc} \; & \sum_{cd}p\left(ab\;\mathrm{at}\;t_{2}|cd\;\mathrm{at}\;t_{1},\rho ,xy\right)p\left(cd\;\mathrm{at}\;t_{1}|\rho ,xy\right)\\ & =\sum_{CD}Tr\left(\rho \sqrt{C}\sqrt{D}U^{†}ABU\sqrt{C}\sqrt{D}\right). \end{array}$$
The summation is over POVM element sets for {C} and {D}. Since $[A⊗I_{L_{2}},I_{L_{1}}⊗B]=0$, the L.H.S. of (22) is $p\left(ab\;\mathrm{at}\;t_{2}|\rho ,xy\right)=Tr\left(\rho U^{†}ABU\right)$. Both sides are not equal in general.

In the case that the post-measurement state after the first measurement is unchanged during free time evolution (this is indeed the assumption in the no-go theorems I will analyze in the next section), Equation (21) becomes
(24)$$p\left(ab\;\mathrm{at}\;t_{2}|cd\;\mathrm{at}\;t_{1},\rho ,xy\right)=\frac{Tr(\rho \sqrt{C}\sqrt{D}AB\sqrt{C}\sqrt{D})}{Tr(\rho CD)}.$$
Equation (23) is simplified to
(25)$$\begin{array}{cc} \; & \sum_{cd}p\left(ab\;\mathrm{at}\;t_{2}|cd\;\mathrm{at}\;t_{1},\rho ,xy\right)p\left(cd\;\mathrm{at}\;t_{1}|\rho ,xy\right)\\ & =\sum_{CD}Tr\left(\rho \sqrt{C}\sqrt{D}AB\sqrt{C}\sqrt{D}\right), \end{array}$$
and $p\left(ab\;\mathrm{at}\;t_{2}|\rho ,xy\right)=Tr\left(\rho AB\right)$. In this case, one can derive the following corollary based on Theorem 1.

**Corollary** **1.** 
*In the Extended Wigner’s Friend scenario setup, suppose the post-measurement state is unchanged during free time evolution from $t_{a}$ to $t_{b}$. Select $t_{1}$ and $t_{2}$ such that $t_{a}<t_{1}<t_{b}<t_{2}$. The law of total probability (22) is true if one of the following conditions is met.*
*C4.* 
*[A,C]=0 and [B,D]=0, $\forall \rho_{0}$,*
*C5.* 
*$\rho_{0}=\left|\Psi \rangle \langle \Psi \right|$, C and D are projection operators, and*

(26)
〈Ψ|[CD,AB]CD|Ψ〉=0.




Condition C4 is quite obvious. Proof of condition (26) is given in Appendix B.

## 4. Logical Loopholes in No-Go Theorems Related to the Wigner’s Friend Scenario

A no-go theorem usually starts from the conventional probability theory, which is widely regarded as the true representation of logical deduction, and assumes certain additional plausible physical premises: realism, locality, no superdeterminism, observer independence, etc. One then shows that such a model leads to prediction, which is contradicted by quantum mechanics. Hence, one concludes that at least one of the assumptions or the rules of conventional probability must be violated by quantum mechanics. Let us denote the physical assumption that a no-go theorem tries to prove to be violated by quantum theory as W. For instance, W could be “measured facts are observer independent”. The no-go theorem may be constructed independent of the underlying physical theory. But if the logical deduction in the proof of theorem utilizes the law of total probability in one of the forms of (12), (16), or (22) without calling out the appropriate sufficient condition *C*, then I know the resulting statement (could be in the form of an inequality) will not hold in quantum theory. This leaves a loophole in the logical deduction. Because the contradiction shown in the no-go theorem could be just due to the fact that *C* is not met in the experiment setup instead of the intended conclusion that W is violated by quantum theory. Thus, the no-go theorem does not reach the conclusion as desired. I will examine several such no-go theorems in this section (I do not include the no-go theorem [33] widely discussed in the literature since its proof does not invoke the law of total probability.)

### 4.1. A Strong No-Go Theorem on the Wigner’s Friend Paradox

Bong et al. introduce a no-go theorem that if one assumes that quantum mechanics is applicable to the scale of an observer, then one of the three assumptions, ‘Locality’, ‘No-superdeterminism’, or ‘Absoluteness of Observed Events (AOE)’ must be false [9]. Here AOE means that “every observed event exists absolutely, not relative to anything or anyone”. The no-go theorem is supposed to be independent of underlying physical theory and is proved in the context of extended Wigner’s friend (EWF) scenario [8], as shown in Figure 1. The measurement results from the friend in the lab can be correlated with the super-observer’s subsequent measurement results. Suppose the measurement outcomes from Alice, Bob, Charlie, and Debbie are a,b,c,d, respectively. Alice can have three different measurement settings, labeled by parameter x∈{1,2,3}. When x=1, Alice opens $L_{1}$ and asks Charlie’s measurement outcome, while when x=2,3, Alice performs a different measurement on $L_{1}$. Similar measurement settings for Bob are labeled as y∈{1,2,3}.

Among the three assumptions, it is well accepted that ‘Locality’ and ‘No- superdeterminism’ cannot be violated by any physical theory. The focus is on the assumption of AOE, which is defined mathematically as following [9]. There exists a joint probability distribution p(abcd|xy) such that

i

$p\left(ab|xy\right)=\sum_{cd}p\left(abcd|xy\right)\forall a,b,x,y$

ii

$p\left(a|cd,x=1,y\right)=\delta_{a,c}\forall a,c,d,y$

iii$p\left(b|cd,x,y=1\right)=\delta_{b,d}\forall b,c,d,x$.

With the three assumptions, [9] derives a number of inequalities and experimentally confirms that quantum theory violates these inequalities when proper measurement settings *x* and *y* and the initial quantum state are chosen. Therefore, the AOE assumption should be refuted.

However, closer examination of the derivation shows that the no-go theorem assumes the law of total probability. The definition of AOE states that $p\left(ab|xy\right)=\sum_{cd}p\left(abcd|xy\right)$. Then, in Equation (3) of [9], it implicitly assumes p(abcd|xy)=p(ab|cdxy)p(cd|xy). Together, they imply
(27)$$p\left(ab|xy\right)=\sum_{cd}p\left(ab|cdxy\right)p\left(cd|xy\right).$$
However, as discussed in Section 2, the law of total probability does not hold true in quantum theory unless a certain condition is met.

We can apply Corollary 1 to analyze the validity of (27). In Appendix C, I show that the operators chosen in [9] are not all commutative with each other. Specifically, [A,C]≠0 and [B,D]≠0, while $[C⊗I_{L_{2}},I_{L_{1}}⊗D]=0$ and $[A⊗I_{L_{2}},I_{L_{1}}⊗B]=0$. With these choices of operators, the corresponding two-time version of the law of total probability is given by (22) and (25). However, I already see condition C4 is not satisfied. The choice of initial quantum state, i.e., Equation (1) in [9] and the forms of operator A,B,C,D do not satisfy conditions C5 in Corollary 1 either.

Therefore, in general, (27) does not hold with the conditions specified in [9]. Inequalities derived based on (27) will be violated by quantum theory with the choice of initial quantum state and measurement operators described in [9]. However, this raises the question of exactly what the no-go theorem refutes. I agree with the authors that violation of the inequalities by quantum theory points to the validity of AOE in quantum theory. However, the definition of AOE and the derivation of the theorem implicitly assume the validity of the law of total probability. The root cause of the violation of the inequalities is due to the fact that the experimental setup does not satisfy the conditions to make the law of total probability hold true, not because of the AOE statement that “an observed event is not relative to anything or anyone”. One may argue that the AOE statement is equivalent to invalidity of the law of total probability. However, as discussed earlier, the invalidity of the law of total probability is due to the fact that measurement *C* (or *D*) alters the initial quantum state that impacts the probability of outcome for measurement *A* (or *B*) since [A,C]≠0 (or [B,D]≠0). There is a logic gap to equate this reason with the statement “an observed event is not relative to anything or anyone”. Therefore, it appears that the violation of inequalities in [9] just reconfirms the consequence of non-commutative measurements and, therefore, the invalidity of the law of total probability in quantum theory, rather than confirming the invalidity of the AOE statement.

### 4.2. A No-Go Theorem for Observer-Independent Facts

The no-go theorem for observer-independent facts by Brukner [8] was actually introduced earlier than [9] in a similar effort to prove that measured facts are observer-dependent. The experimental setup is shown in Figure 1. In [8], there are only two different measurement setups either Alice or Bob will perform, compared to three different measurement setups in [9]. Furthermore, the way the no-go theorem is proven is different. The proof in [8] leverages the well-known Bell–CHSH inequality, and thus inherits all the assumptions associated with the Bell–CHSH theorem, while the no-go theorem in [9] is proven independent of the Bell–CHSH theorem. As I will explain next, the proof in [8] is more subtle, as it carefully chooses a quantum state and a set of measurement operators such that the law of total probability holds true if only considering Alice’s (or Bob’s) measurements.

The initial wave function is chosen such that after Charlie and Debbie each perform a measurement of their respective half of entangled spin along the *z* axis, from Alice’s or Bob’s perspective, it becomes (see Equations (5)–(9) in [8]):(28)$$\begin{array}{cc} |\Psi \rangle & =-\frac{1}{\sqrt{2}}\sin\frac{\theta }{2}{(|00\rangle }_{L_{1}}{|00\rangle }_{L_{2}}+{|11\rangle }_{L_{1}}{|11\rangle }_{L_{2}})\\ \; & +\frac{1}{\sqrt{2}}\cos\frac{\theta }{2}{(|00\rangle }_{L_{1}}{|11\rangle }_{L_{2}}-{|11\rangle }_{L_{1}}{|00\rangle }_{L_{2}}), \end{array}$$
where ${|00\rangle }_{L_{2}}$ represents that $s_{1}$ is in the spin up state and Charlie’s pointer variable is associated with the up state, and ${|11\rangle }_{L_{1}}$ corresponds to the spin down state for $s_{1}$ and Charlie’s pointer variable. There are similar meanings for ${|00\rangle }_{L_{2}}$ and ${|11\rangle }_{L_{2}}$ for $s_{2}$ and Debbie’s pointer variable. The key point of [8] is to assume there exists a joint probability $p(a_{1}a_{2}b_{1}b_{2})$, where $a_{1},a_{2}\in \left\{0,1\right\}$ are the measurement results corresponding to Alice’s choice of two types of measurement operations, and $b_{1},b_{2}\in \left\{0,1\right\}$ are the measurement results of Bob. Alice can choose to either measure $L_{1}$ with projection operator $A_{1}$ or with projection operator $A_{2}$. Here $A_{1}$ is represented (in [8], the two types of operations for Alice are defined as $A_{1}={\left|00\rangle \langle 00\right|}_{L_{1}}-{\left|11\rangle \langle 11\right|}_{L_{1}},i\in \left\{0,1\right\}$ and $A_{2}={\left|00\rangle \langle 11\right|}_{L_{1}}-{\left|00\rangle \langle 11\right|}_{L_{1}}$. Here I use the spectral decomposition theorem to decompose $A_{1}$ into projection operators and represent it by $A_{1}$, with a similar approach for the definition of $A_{2}$. There is an important difference here compared to the setup in [9]. Here $a_{1},a_{2}$ are results for Alice from completed measurement $A_{1},A_{2}$, respectively, whereas in [9], a,c are measurement results for Alice and Charlie, respectively. The issue of *c* as a result of Charlie’s “pre-measurement” in [9] does not exist here in [8]) by $|\phi_{i}\rangle \langle \phi_{i}{|}_{L_{1}}$, and $|\phi_{0}\rangle =|00\rangle$ for $a_{1}=0$ or $|\phi_{1}\rangle =|11\rangle$ for $a_{1}=1$. $A_{2}$ is chosen to be $A_{2}=|\chi_{0}\rangle \langle \chi_{0}|$ for $a_{2}=0$ or $A_{2}=|\chi_{1}\rangle \langle \chi_{1}|$ for $a_{2}=1$, where
(29)$$|\chi_{i}\rangle =\frac{1}{\sqrt{2}}{({\left(-1\right)}^{i}|00\rangle }_{L_{1}}+{|11\rangle }_{L_{1}}).$$
From these definitions of $A_{1}$ and $A_{2}$, one can verify that
(30)$$\left[A_{1},A_{2}\right]={\left(-1\right)}^{a_{1}+a_{2}}(|00\rangle {\langle 11|}_{L_{1}}-|11\rangle \langle 00{|}_{L_{1}}),$$
with similar definitions for operators $B_{1}$ and $B_{2}$. The problem in [8] is that it assumes the law of marginal probability holds true, for instance, $p\left(a_{2}b_{2}\right)=\sum_{a_{1},b_{1}=\left\{0,1\right\}}p\left(a_{1}a_{2}b_{1}b_{2}\right)$. Ref. [8] does not provide details on how the joint probability is defined. As discussed earlier, the joint probability cannot be well defined unless the measurement operators are commutative. If I further assume the validity of the classical probability axiom in Equation (1) and apply it recursively, I have
(31)$$\begin{array}{cc} p(a_{1}a_{2}b_{1}b_{2}) & =p\left(a_{2}b_{1}b_{2}|a_{1}\right)p\left(a_{1}\right)\\ \; & =p\left(a_{2}b_{2}|a_{1}b_{1}\right)p\left(b_{1}|a_{1}\right)p\left(a_{1}\right)\\ \; & =p\left(a_{2}b_{2}|a_{1}b_{1}\right)p\left(a_{1}b_{1}\right). \end{array}$$
Then, the law of marginal probability is equivalent to the law of total probability such that
(32)$$p\left(a_{2}b_{2}\right)=\sum_{a_{1},b_{1}=\left\{0,1\right\}}p\left(a_{2}b_{2}|a_{1}b_{1},\rho \right)p\left(a_{1}b_{1}\right).$$
Let us analyze if Equations (31) and (32) hold true with the chosen operators $\left\{A_{1},B_{1},A_{2},B_{2}\right\}$ and the quantum state in (28). To do this, I apply Corollary 1 by replacing operators {A,B,C,D} with operators $\left\{A_{2},B_{2},A_{1},B_{1}\right\}$, respectively, and setting $\rho_{0}=\left|\Psi \rangle \langle \Psi \right|$, where |Ψ〉 is defined in (28). The conditional probability is similar to (24),
(33)$$\begin{array}{cc} \; & p(a_{2}b_{2}\;\mathrm{at}\;t_{2}|a_{1}b_{1}\;\mathrm{at}\;t_{1},\rho ,xy)\\ & =\frac{Tr(\rho \sqrt{A_{1}}\sqrt{B_{1}}A_{2}B_{2}\sqrt{A_{1}}\sqrt{B_{1}})}{Tr(\rho A_{1}B_{1})}, \end{array}$$
where I drop the subscript of $\rho_{0}$. The desired law of total probability is
(34)$$\begin{array}{cc} \; & p(a_{2}b_{2}\;\mathrm{at}\;t_{2}|\rho )=\\ \; & \sum_{a_{1},b_{1}}p\left(a_{2}b_{2}\;\mathrm{at}\;t_{2}|a_{1}b_{1}\;\mathrm{at}\;t_{1},\rho \right)p\left(a_{1}b_{1}\;\mathrm{at}\;t_{1}|\rho \right). \end{array}$$
In Appendix D, I show that for the choices of the set of operators $\left\{A_{1},A_{2},B_{1},B_{2}\right\}$ prescribed earlier, $[A_{1},A_{2}]\neq 0$ and $[B_{1},B_{2}]\neq 0$. With the quantum state (28), Condition $C3^{\prime }$ is satisfied such that $p\left(a_{2}|\rho \right)=\sum_{a_{1}}p\left(a_{2}|a_{1},\rho \right)p\left(a_{1}|\rho \right)$ and $p\left(b_{2}|\rho \right)=\sum_{b_{1}}p\left(b_{2}|b_{1},\rho \right)p\left(b_{1}|\rho \right)$. Unfortunately, I also show that (34) and the law of marginal probability $p\left(a_{2}b_{2}\right)=\sum_{a_{1},b_{1}=\left\{0,1\right\}}$$p(a_{1}a_{2}b_{1}b_{2})$ are still not valid.

Since proof of the no-go theorem in [8] depends on the law of marginal probability, and the law of marginal probability does not hold true by the choice of quantum state and measurement operators, there is a logical loophole in the no-go theorem. The violation of the inequality in [8] in quantum theory does not necessarily imply that measured facts are observer dependent. Instead, the violation just reconfirms that the law of marginal probability does not hold for the choice of quantum state and the measurement operators. The logical gap of equating the statement “measured facts are observer dependent” to the invalidity of the law of marginal probability in quantum theory is similar to what I discussed in the last paragraph in Section 4.1.

Note that besides depending on the law of marginal probability, the proof of no-go theorem in [8] also inherits the assumptions for the proof of the Bell–CHSH inequality [31,32], particularly dependency on the outcome independence assumption (19). The no-go theorem in [9], on the other hand, does not depend on the outcome independence assumption.

### 4.3. A No-Go Theorem for the Persistent Reality of Wigner’s Friend’s Perception

In [11], another no-go theorem is introduced to show that in the extended Wigner’s friend scenario, Wigner’s friend cannot “treat her perceived measurement outcome as having reality across multiple times” without contradicting one of the following assumptions in quantum mechanics [11].

P1Let $f_{1}$ and $f_{2}$ be perceived measurement records of the friend at time $t_{1}$ and $t_{2}$, respectively. A joint probability distribution $p(f_{1},f_{2})$ can be assigned that also satisfies the law of total probability $p\left(f_{1}\right)=\sum_{f_{2}}p\left(f_{1},f_{2}\right)$ and $p\left(f_{2}\right)=\sum_{f_{1}}p\left(f_{1},f_{2}\right)$;P2One time probability is assigned according to $p\left(f_{i}\right)=Tr(|f_{i}\rangle \langle f_{i}|\rho )$ using unitary quantum theory where no state collapse is considered to have occurred;P3The joint probability of the friend’s perceived outcomes $p(f_{1},f_{2})$ has a convex linear dependence on the initial state ρ.

In traditional quantum measurement theory [30], the unitary process is considered to entangle the measured system with the measuring apparatus before the projection process. The projection process gives a definite final outcome. P2 essentially assumes the unitary process itself can have a measurement result and can be assigned a (one-time) probability. Zukowski and Markiewicz have already pointed out that such an assumption leads to a contradiction. However, there is another problem with P2 [26]. The derivation in [11] assumes that the joint probability $p(f_{1},f_{2})$ is derived through the standard probability axiom $p\left(f_{1},f_{2}\right)=p\left(f_{2}|f_{1}\right)p\left(f_{1}\right)$, but it does not give details on how the conditional probability is calculated in quantum theory. It is not clear how the unitary formulation presented in [11] can be applied to derive the conditional probability $p(f_{2}|f_{1})$ because P2 assumes there is no “collapse” after the first measurement. It is not a problem to compute the one-time probability $p(f_{1})$ and $p(f_{2})$. However, in order to be able to calculate a two-time probability such as $p(f_{2}|f_{1})$, one will have to apply the state update rule after the first measurement at time $t_{1}$, as shown in (13) for the two-time conditional probability.

More crucially, even if I am able to calculate the conditional probability, there is still a problem with P1, as P1 assumes the law of total probability $p\left(f_{1}\right)=\sum_{f_{2}}p\left(f_{1},f_{2}\right)$ is always true. We have shown in Theorem 1 and subsequent corollaries that the law of total probability is true in quantum theory only with certain conditions. The two POVM elements chosen in [11] are non-commutative, as shown in Equation (17) in [11]. Thus, $p\left(f_{1}\right)=\sum_{f_{2}}p\left(f_{1},f_{2}\right)$ does not necessarily hold. The proof in [11] assumes that $p\left(f_{1}\right)=\sum_{f_{2}}p\left(f_{1},f_{2}\right)$ always holds based on P1, then eventually deduces that the two POVM elements should be commutative and claims there is a contradiction. However, such a contradiction is due to the invalid assumption of $p\left(f_{1}\right)=\sum_{f_{2}}p\left(f_{1},f_{2}\right)$ in P1, which in turn is due to the fact that the two POVM elements are non-commutative. Since P1 is invalid, the contradiction does not lead to the desired conclusion that Wigner’s friend cannot “treat her perceived measurement outcome as having reality across multiple times”.

### 4.4. Relative Facts, Stable Facts

In relational interpretation of quantum mechanics (RQM) [34,35,36,37,38,39], a measurement result is considered meaningful only relative to the system that interacts with the measured system. A definite measurement result is referred to as a fact. Quantum theory is about conditional probability for facts, given other facts. Recently, Biagio and Rovelli introduced the concept of a *stable fact* in the following sense [10]. If, given the probability $p(a_{i})$ for *N* mutual exclusive facts $a_{i}\left(i=1\ldots N\right)$ and the conditional probability of another fact *b*, $p(b|a_{i})$, the probability p(b) (dropping index *j* for $b_{j}$) is given by (4), then facts $a_{i}$ are considered stable.

RQM states that fact is relative. Formally, if two systems *S* and *F* interact such that variable $L_{F}$ of *F* depends on the value of variable $L_{S}$ of *S*, then the value of $L_{S}$ is said to be relative to *F* [10]. However, not all relative facts are stable. The main thesis of [10] consists two claims. First, the law of total probability (4) is satisfied only if *b* and $a_{i}$ are facts relative to the *same* system, say relative to system *F*. If *b* is relative to another system W≠F, (4) is not true in general. Mathematically, these can be expressed as
(35)$$\begin{array}{cc} p(b^{\left(F\right)}) & =\sum_{i}p\left(b^{\left(F\right)}|a_{i}^{\left(F\right)}\right)p\left(a_{i}^{\left(F\right)}\right) \end{array}$$
(36)$$\begin{array}{cc} p(b^{\left(W\right)}) & \neq \sum_{i}p\left(b^{\left(W\right)}|a_{i}^{\left(F\right)}\right)p\left(a_{i}^{\left(F\right)}\right). \end{array}$$
Here, it is important to label the reference system the fact is relative to. Second, if system *F* goes through a decoherence process by interacting with an environmental system *E*, the resulting density matrix for *F* is approximately given by $\rho =\sum_{i}\lambda_{i}|Fa_{i}\rangle \langle Fa_{i}|$, where $Fa_{i}$ is the eigenvalues of $L_{F}$. Then, Equation (36) can be rewritten as
(37)$$p\left(b^{\left(W\right)}\right)=\sum_{i}p\left(b^{\left(W\right)}|Fa_{i}^{\left(E\right)}\right)p\left(Fa_{i}^{\left(E\right)}\right).$$
In such a case, facts $Fa_{i}$ relative to *E* are stable for *W*.

Now let us examine the two claims more carefully. For the first claim, from Theorem 1, Equation (35) is not necessarily true even if both *b* and $a_{i}$ are facts relative to a same system. That facts *b* and $a_{i}$ are both relative to a same system means both facts are obtained through interactions with the same system, and the interactions can be represented by measurement operators *B* and $A_{i}$, respectively. If $[B,A_{i}]=0$, Equation (35) is true. But there is no reason that *B* and $A_{i}$ have to be commutative. If $[B,A_{i}]\neq 0$, Equation (35) is not true in general, unless other conditions such as condition $C2^{\prime }$ or $C3^{\prime }$ in Theorem 1 are satisfied. Indeed, the second claim (37) is precisely the case where condition $C2^{\prime }$ is met. Note that the reasoning from (35) to (37) is also applicable when facts *b* and $a_{i}$ are relative to the same system but the corresponding measurement operators are non-commutative, $[B,A_{i}]\neq 0$.

Therefore, it is not clear that one can use the validity of the law of total probability (35) and (37) to distinguish stable facts from non-stable facts. Again, I am not opposed to the idea that facts are relative. What I am questioning here is the validity of (35) without specifying the conditions, and the rigorousness of reasoning from (36) to (37). It appears that more careful investigation is needed in order to search for the criteria to define a “stable” fact.

## 5. Discussion and Conclusions

### 5.1. The Page–Wootters Timeless Formulation

In the timeless formulation of quantum theory developed by Page and Wootters [21], time evolution is naturally emerged from quantum correlation between a clock and a system whose dynamics are tracked by the clock. Ref. [25] proposed several two-time formulations of conditional probability based on the Page–Wootters timeless mechanism. The advantage of such formulation is that from a timeless quantum state one can derive probability of a measurement event conditional on another event regardless of the temporary order of the two events.

Although the formulation in the present work is based on the regular time evolution dynamics in the Schrodinger picture, the definition of two-time conditional probability (13) is consistent with the definitions in [25]. For instance, for the case of two projection measurements $A_{i}$ and $B_{j}$ at $t_{a}$ and $t_{b}$, respectively, (13) gives the same transition probability (14) as that in Equation (29) of [25].

However, the timeless formulations of conditional probability in [25] are applicable only to projection measurements, while the theory developed here is more general in the sense that it is applicable to POVM measurements. A two-time conditional probability formulation for projection measurements is insufficient to analyze the no-go theorems in [9]. Moreover, my focus here is the validity of the law of total probability that is built on the definition of two-time conditional probability, which is missing in [25], as the focus there is only on the rules for two-time conditional probability.

It will be interesting to generalize the timeless Page–Wootters formulation of two-time conditional probability in [25] to be able to handle POVM measurements, although I expect such generalization should produce results similar to those presented in this work.

### 5.2. Limitations

One limitation of the present work is that in Theorem 1, I am only able to derive three sufficient conditions for the law of total probability to hold true. In theory, there can be many other sufficient conditions. It is desirable to find the *sufficient and necessary* condition for the law of total probability to hold true in quantum theory. This remains a future investigation topic. Nevertheless, for the purpose of analyzing the EWF scenario and identifying the loopholes of the relevant no-go theorems, the conditions specified in Theorem 1 and subsequent corollaries are sufficient.

### 5.3. Conclusions

In this paper, the standard rule to assign conditional probability in quantum theory, i.e., Lüders rule, is extended to include two-time POVM measurements. The extension is strictly based on the recursive application of the POVM measurement theory as shown in (5) and (6) and the assumption that probability distribution can be assigned only for completed quantum measurement. The resulting definition (13) is consistent with other works based on Page–Wootters formulation [25], but with advantage of being able to apply to POVM measurements instead of just projection measurements.

More importantly, with the generalized two-time conditional probability formulation, I analyze the validity of the law of total probability. It is shown that the quantum version of the law of total probability does not hold true in general. Certain conditions related to the choice of measurement operators and the initial quantum state must be met in order for the law of total probability to hold. Specifically, such sufficient conditions are derived in Theorem 1 and Corollary 1.

Application of the theory developed here to the extended Wigner’s friend scenario reveals logical loopholes in several no-go theorems. These no-go theorems take for granted the validity of the law of total probability (or the law of marginal probability) in quantum theory. However, this is not the case, as shown in Theorem 1 and Corollary 1. Thus, the no-go theorems do not lead to the desired conclusions. For instance, the violation of the inequalities developed in [8,9] in quantum theory does not necessarily lead to the desired statement that “measured facts are observer-dependent”. Instead, it just reconfirms the invalidity of the law of total probability or the law of marginal probability in quantum theory. I do not take a stand on the assertions themselves of the no-go theorems. It could be still a valid statement that “measured facts are observer-dependent”. What I show here is that there are logical loopholes to reach such a statement. It is desirable to find more convincing proof and experimental testing because the implications of the extended Wigner’s friend scenario are conceptually fundamental in quantum theory.

## Figures and Tables

**Figure 1 entropy-24-00903-f001:**
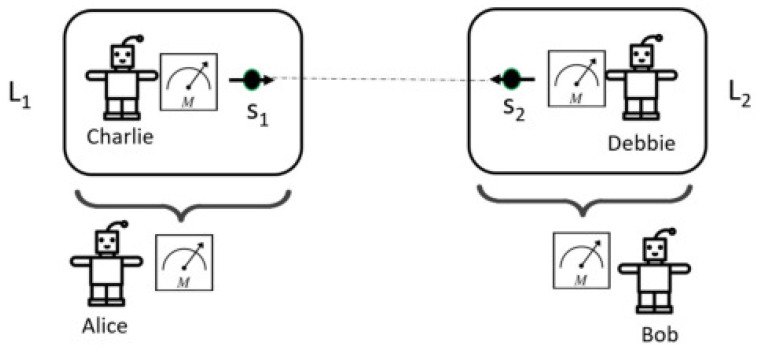
Sketch of the extended Wigner’s friend scenario described in [8]. Laboratory $L_{1}$ consists of spin $s_{1}$ and Charlie, while Laboratory $L_{2}$ consists of spin $s_{2}$ and Debbie. The two laboratories are remotely separated. The dotted line between $s_{1}$ and $s_{2}$ symbolizes they are entangled. Alice can measure $L_{1}$ as a whole, and Bob can measure $L_{2}$.

## Data Availability

The data that support the findings of this study are available within the article.

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
