# Peer review of "Law of Total Probability in Quantum Theory and Its Application in Wigner’s Friend Scenario"

_entropy, 2022, doi:10.3390/e24070903_

Round 1

Reviewer 1 Report

The authors provide a re-analysis of the validity of the law of total probability in the context of quantum mechanics. They generalize the law from the usual projective measurements to two-time POVMs (positive operator-valued measurements), which enables them to re-analyze recently proposed no-go theorems which aim at supporting the claim that “measured facts are observer-dependent”, and which invoke extended versions of the famous Wigner's friend scenario. The authors conclude that the re-analyzed no-go theorems each contain a loophole based on a faulty application of the law of total probability, which undermines the general validity of the no-go theorems and their support for the claimed observer-dependence of measured facts. The study uses basic quantum mechanical algebra and measurement theory, and is straightforward in its argumentation, which I find convincing. Indeed, the loophole that the authors spotted in the published studies seems to me rather glaring, and I am astonished that it had not been uncovered before, already in the course of the respective review processes. The manuscript is generally well-written and structured and I did not find relevant flaws in the mathematics or the argumentation. I only have some minor criticism which should be addressed:
  1. The formulation “instead of due to” is used at several places in the paper, and it seems wrong to me, or weird at least, although I'm not a native speaker. I recommend finding a better formulation, e.g. in the Abstract “Consequently, the contradictions in these no-go theorems just reconfirm the invalidity of the law of total probability in quantum theorem, instead of due to the physical statements that the no-go theorems attempt to refute” could be replaced by “ ... rather than confirming the physical statements ...”.
  2. When citing, please cite authors or studies, not just numbers. E.g. line 103 “[7] gives an extensive review and discussion” could be replaced by “G. Bobo gives an extensive review and discussion [7]”. Similarly, “[15] derives a number of inequalities” should rather become “Bong et al. derive a number of inequalities [15]”.
  3. As for Ref. 7, these are actually two different publications, which should be listed separately.
  4. Occasional illegal use of the definite article, e.g. in LL 105-106: “The Lüders rule is based on the Gleason’s theorem which mathematically justifies the Born’s rule” can be stripped of all instances of “the”.
  5. Would the main theorem not be even more elegant using Kraus operators for the POVM? The authors stick to square root operators, e.g. in Eq. 6, when representing the post-measurement state. They then add unitary time evolution U, which yields their condition C1 in line 141. Using Kraus operators, which need not be self-adjoint, the combination of the square-root operators sqrt(A) and sqrt(B) with the two-time unitary time evolution operator U would be represented by one Kraus operator each, so instead of sqrt(A) U they could directly use non-self-adjoint Kraus operators A and B with the POVM elements then being E = A^+ A and F = B^+ B. But this is just a suggestion.
  6. The theorems and corollaries should be explicitly indicated. The authors refer to “Theorem 2” and “Corollary 2”, but where are they? I can only gather from the context that “Theorem 2” starts in line 136 and ends in line 144. But where is “Theorem 1” and “Corollary 1”? Please represent theorems and corollaries in a complete and explicit fashion, in particular indicate start and end of each theorem and provide the number.
  7. Is their “Corollary 2” really necessary? It is completely straightforward that when the post-measurement state is unchanged, hence when the unitary time evolution operator acts as unity on the post-measurement state, that then C4 follows from C1, that is, that A and B must commute. I would recommend removing that Corollary altogether, which also makes it unnecessary to provide differently numbered conditions C4-C6, from which only C4 differs from the corresponding condition in the main theorem. My suggestion would be: Give your theorem a more formal presentation, maybe use Kraus operators (see above) to put it as elegant as possible, and do without Corollary 2.
  8. P. 4: “Strictly speaking, when Ai and Bj are non-commutative, due to the uncertainty principle, there are no definite outcomes of the two measurements are performed at the same time.” The sentence is grammatically incomplete and semantically unclear. When Ai and Bj do not commute, the respective measurements can in principle not be performed at the same time anyway. This should be clarified.
  9. P. 4: “(7) or (8) is then considered as a special case when [Ai, Bj] = 0 and the measurement times coincide.” First, avoid starting a sentence with a number. Here, we can put it as “Eq. (7) or (8)...” Second, I do not agree with the statement here. For non-commuting operators A and B, equations (7) and (8) are simply not identical, that is all. The two equations are generally identical if and only if the operators commute, in which case, and in which case only, the measurement times can coincide. 
  10. Page 11: You mention “classical probability axiom 1” without reference in the paper. The probability axioms are certainly well-known, but their numbering is arbitrary, so I would recommend stripping the number here.
  11. Replace “underline” with “underlying” at several places. I also found some more typos, so I recommend proofreading the manuscript by a competent speaker.
With these issues adequately addressed I recommend the publication of the manuscript.

Reviewer 2 Report

This paper describes theoretical work which generalises previous discussions of the Wigner’s Friend scenario to include the measurement of POVMs.   It is clearly written, the results appear to be sound and the paper makes a significant contribution to the understanding of the conceptual basis of quantum mechanics.  I recommend acceptance.

Author Response

I sincerely thank the referee for carefully read through the manuscript and recommendation of acceptance. An acknowledgement to all the referees is added in the end of the revised manuscript.

Round 2

Reviewer 1 Report

The author has suitably addressed all my criticisms one by one. However, there still remains some minor issues to be addressed
  1. In the abstract, contrary to my own suggestion, and I apologize for that, the modified sentence is now indeed stating the opposite of what it was supposed to state. I suggest the following formulation: 
    Consequently, the contradictions in these no-go theorems only reconfirm the invalidity of the law of total probability in quantum theory, rather than invalidating the physical statements that the no-go theorems are intended to refute.
  2. The same applies to a variation of the statement on page 2, where I suggest the following formulation: 
    The contradictions in these no-go theorems only reconfirm the invalidity of the law of total probability in quantum theory, rather than invalidating the physical statements that the no-go theorems are intended to refute, such as “measured facts are independent of the observer . 
    Or, alternatively: 
    The contradictions in these no-go theorems only reconfirm the invalidity of the law of total probability in quantum theory, rather than supporting the physical statements that the no-go theorems are intended to support, such as “measured facts are observer independent”.
  3. Page 4: “With such special post-measurement quantum state, the sufficient conditions in Theorem 1 become” (instead of “becomes”, because the referent of the verb is the sufficient conditions, which is plural).
  4. Page 4: I would suggest to re-label the sufficient conditions, because they are analog but not equal to those above, as C1', C2', C3' (with a prime), so that they can uniquely be referenced later on.
  5. Equations 21, 22, and so on, as well as in the text: It is hard to directly recognize, in a mathematical formula, the string “ab at t_2” in such a way that “ab” are two variables, and “at” is one English word, as both are italic and look very similar. For these cases Latex has the “\text” command, which causes its argument to be written in a same format as text strings in a mathematical context. Specifically, it would cause “at” be not italic, and therefore easier to distinguish from two variables “ab”. 
  6. Is Corollary 1.1 true “if” one of the following conditions apply, or is it true “if and only if at least one” of the following conditions apply? I suppose the latter, but I may be wrong. In any case, this should be clarified, as it potentially makes the Corollary stronger.
  7. Page 7: It's “Bong et al.” rather than “Bong, et al.”
  8. Page 8: Actually, the axiom of classical probability is not formulated with four variables but two, namely “P(A,B) = P(A|B)P(B)”. Of course, the expression used by the authors follows directly by substituting A and B with (a_1, a_2) and (b_1, b_2), respectively. Anyway, however, the axiom is not literally the one stated by the authors. I would prefer a formulation such as 
    If we assume that p(a1a2b1b2) = p(a2b2|a1b1)p(a1b1), which can directly be inferred from the axiom P(A,B) = P(A|B)P(B) of classical probability theory, then ...
  9. Sometimes there is a missing space between “Ref.” and the bracketed number, as in “Ref.[20]”, which should be “Ref. [20]”.
